# Optimal marker gene selection for cell type discrimination in single cell analyses

Bianca Dumitrascu[1,7], Soledad Villar[2,3,7], Dustin G. Mixon[4] & Barbara E. Engelhardt [5,6✉]

Single-cell technologies characterize complex cell populations across multiple data modalities at unprecedented scale and resolution. Multi-omic data for single cell gene expression, in situ hybridization, or single cell chromatin states are increasingly available across diverse tissue types. When isolating specific cell types from a sample of disassociated cells or performing in situ sequencing in collections of heterogeneous cells, one challenging task is to select a small set of informative markers that robustly enable the identification and discrimination of specific cell types or cell states as precisely as possible. Given single cell RNA-seq data and a set of cellular labels to discriminate, scGeneFit selects gene markers that jointly optimize cell label recovery using label-aware compressive classification methods. This results in a substantially more robust and less redundant set of markers than existing methods, most of which identify markers that separate each cell label from the rest. When applied to a data set given a hierarchy of cell types as labels, the markers found by our method improves the recovery of the cell type hierarchy with fewer markers than existing methods using a computationally efficient and principled optimization.

[1] Department of Computer Science and Technology, University of Cambridge, Cambridge, UK. [2] Department of Applied Mathematics and Statistics, Johns Hopkins University, Baltimore, MD, USA. [3] Mathematical Institute for Data Science, Johns Hopkins University, Baltimore, MD, USA. [4] Department of Mathematics, The Ohio State University, Columbus, OH, USA. [5] Department of Computer Science, Princeton University, Princeton, NJ, USA. [6] Center for Statistics and Machine Learning, Princeton University, Princeton, NJ, USA. [7] These authors contributed equally: Bianca Dumitrascu, Soledad Villar.
✉email: bee@princeton.edu

Single-cell RNA-seq (scRNA-seq) has generated a wealth of data allowing researchers to measure and quantify RNA levels in single cells at unprecedented scales[1,2]. These studies yield valuable insights regarding intrinsic properties of single cells and specific cell types, which is critical to understanding cell development and disease[3]. When coupled with other data modalities, such as measurements of cell surface protein levels[4] or spatial transcriptomics[5], a more precise and complex partition of the cell type landscape emerges.

Three single-cell methodologies motivate our work. First, scRNA-seq performs short read RNA sequencing on disassociated cells in a sample; a key goal of scRNA-seq analyses is to label each of the cells in the sample with a precise cell type by considering the genes that are expressed in the cell. Second, single-molecule fluorescence in situ hybridization (smFISH) approaches assay transcriptional data spatially[6]. These techniques rely on fluorescent probes that bind near genes of interest to be quantified called markers. When the marker genes bound by probes are expressed above a threshold, fluorescence can be detected using microscopy at the location of expression. Third, sorting techniques such as fluorescence-activated cell sorting rely on a small number of cell surface marker probes that can accurately distinguish cell types according to differences in expression of a small set of cell surface markers[7]. When a cell passes through a flow cytometer to be sorted, the cell is labeled rapidly based on these markers and sorted appropriately.

If we are to use these methods to visualize, characterize, and distinguish cell labels among collections of heterogeneous cells, a key challenge from these emerging technologies is to label each assayed cell with a precise cell type or cell state label. To do this, gene panels must be designed that allow us to distinguish between cell labels both efficiently (with as few markers as possible) and at high precision (discriminating similar cells with different labels). The number of markers is experimentally constrained by the product of the number of fluorescence channels and the number of hybridization cycles[5]. In particular, state-of-the-art smFISH methods use on the order of 40 markers. The task of optimally choosing markers among all genes that most reliably and precisely distinguish cell labels given a hierarchical partitioning of cell type labels is a combinatorially difficult problem.

Existing supervised approaches to marker selection are scarce and only allow for the identification of markers that distinguish each single-cell labels from all of the other cell labels in a sample (one-vs-all)[8-10]. These methods identify markers that are differentially expressed across two groups by comparing within-group expression with across-group expression. One such method, COMET[10], exhaustively select sets of $k$ markers that it then ranks in a one-vs-all task. This has complexity $G^k$, where $G$ is the number of genes, and it quickly becomes infeasible as $k$ reaches even 4. In fact, COMET proposes at most four gene panels, thus lacking both scalability and generalizability. Furthermore, these approaches ignore both hierarchical relationships among cell labels and correlations in expression patterns among genes. The simplistic one-vs-all representation of cell labels to select markers prevents a solution to the problem when the number of cell labels is larger than the number of markers that can be used in an experiment[11].

In contrast, label hierarchy-aware approaches have the ability to select markers that partition the labels at layers that are not exclusively at the leaves of the hierarchy, allowing genes that are robustly differentially expressed across a subset of cell labels to be selected as markers. For a bifurcating tree representation of the cell type hierarchy, the number of markers required is $k$-1, where $k$ is the number of cell labels (i.e., leaves in the hierarchical tree). Yet, these markers will be redundant if the latent dimension of the space spanned by the gene expression profiles of each cell type is lower than the number of cell labels we aim to distinguish. Assuming this is true, then a number of markers smaller than $k$ is needed to maintain the hierarchy.

To fill this methodological gap, we developed scGeneFit, a rigorous and efficient approach for marker selection in the context of scRNA-seq data with a given hierarchical partition of labels. Our method draws from ideas in compressive classification[11] and largest margin nearest neighbor algorithms[12,13]. scGeneFit shows good performance in both simulated data and in scRNA-seq data. Where traditional approaches test the discriminatory value of each gene separately, our approach jointly recovers the optimal set of genes of a given size that allows robust partitioning of the given labels. Our framework generalizes to settings where the input label partition is captured in a hierarchical, or tree-like, structure.

## Results

Briefly, scGeneFit works as follows. Given samples (cells) in a high-dimensional feature space (genes), and corresponding categorical sample labels (e.g., cell type and cell state), label-aware compression methods[13] find a projection to a low-dimensional subspace, or the space of selected markers, where the dimension is specified. Samples with the same labels are closer in the low-dimensional space than samples with different labels, when projected into the low-dimensional space. To ensure that the low-dimensional results enable marker selection, scGeneFit additionally constrains the projection so that each of the subspace dimensions must be aligned with a coordinate axis in the original space. With this constraint, each dimension in the low-dimensional space—capturing a single marker—corresponds to a single gene, and not a weighted linear combination of many genes. Fortunately, this constraint eases the computational challenge of the original projection problem, which is intractable in its most general form. This optimization becomes a linear program.

For input to our method, we use post-quality control scRNA-seq data with unique molecular identifier counts, a target marker set size, and a hierarchical taxonomy of cell labels. When cell labels do not exist, labels may be inferred using a clustering algorithm, or via another data modality[4]. Similarly, the label hierarchy for input to our method can be expert-provided[14], or inferred via a hierarchical clustering algorithm. We assessed scGeneFit's performance in both simulated scenarios and existing scRNA-seq data.

**Spectral toy model.** We first investigated the behavior of scGeneFit in the context of simulated data that is often used to evaluate methods for spectral clustering. Samples of dimension $d + 2$ were generated such that $d$ features are each drawn from a Gaussian distribution with mean zero and variance $\sigma = 1$, and two features, encoding the desired clusters, represent the two-dimensional coordinates on one of two concentric circles with different radii; the cluster label represents whether the sample is on the inner or the outer circle (Fig. 1A; see "Methods"). We found that scGeneFit selected as markers the two features representing the circle coordinates whenever the size of the target marker set was ≥2, enabling the recovery of the sample labels. In contrast, methods that query each of the features independently are not able to identify the two-dimensional coordinates.

**Heteroskedastic toy model.** To illustrate that scGeneFit can correctly identify multiple planted labels within the data, we simulated data corresponding to fantasy gene expression profiles from $n = 1000$ cells and $d = 10,000$ genes (Fig. 1B). We considered three different labelings of these $n$ cells and assigned 250 genes as markers of each of three partitions and another 9250

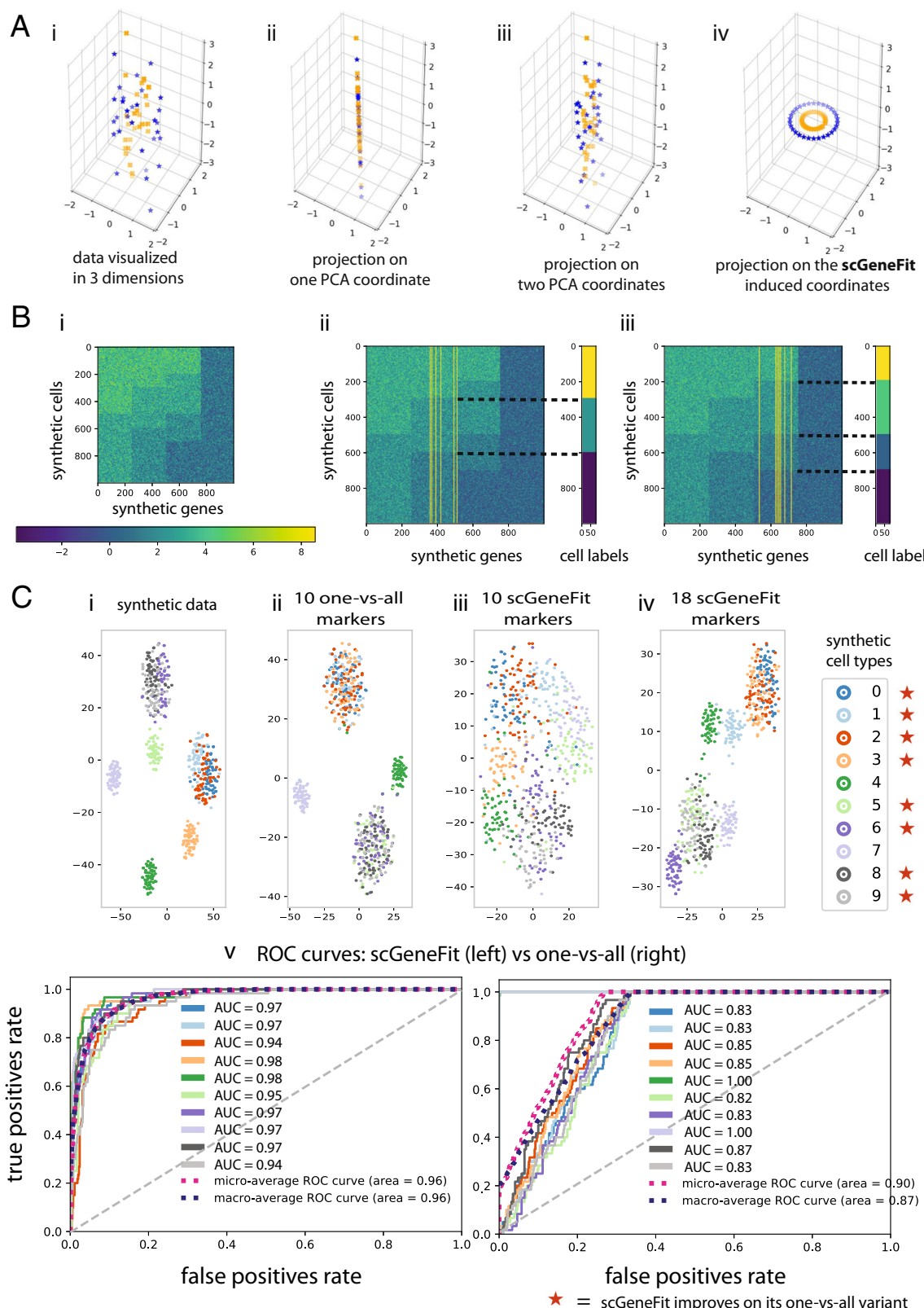

genes as non-markers. Within each of the three labelings, the 250 markers for each label were drawn from a multivariate Gaussian with a 250 dimensional mean (a random sample with mean 0 and variance 1), and a dense $250 \times 250$ covariance matrix. The non-marker genes were drawn from univariate Gaussian distributions

with mean 0 and variance 1. Since the class labels depend on the covariance structure, no individual group of genes considering either mean or variance is predictive of class label. We set the marker set size to 10 and include as input one of two labelings. scGeneFit correctly recovers markers of the appropriate labeling.

**Fig. 1 scGeneFit identifies markers associated with a flat partition of cell type labels when applied to a wide range of synthetic datasets. A** Proof of concept inspired by ref. [13]; cells are color coded with labels. In simulated high-dimensional data, for each sample, two dimensions (x- and y-axes) are drawn from concentric circles, and the remaining dimensions are drawn from white noise. The underlying structure is not apparent from the data (**A**-i). Considering each dimension in isolation, marker selection fails to capture the true structure (**A**-ii,iii). In contrast, scGeneFit recovers the correct dimensions as markers, and is able to recapitulate the label structure (**A**-iv). **B** Discriminative markers were correctly recovered by scGeneFit for simulated samples drawn from mixtures of Gaussians corresponding to two distinct label sets with three (**B**-ii), and four (**B**-iii) labels, respectively. Each row is a single sample and each column is a single feature or gene. Only 1000 genes of 10,000 are visualized, representing all the types simulated. The yellow lines correspond to the markers selected by scGeneFit. **C** t-SNE visualizations of results from the functional group synthetic data (**C**-i–iv). ROC curves comparing the performance of one-vs-all and scGeneFit in distinguishing cell labels following dimension reduction. scGeneFit outperforms one-vs-all in most cell labels when using the same number of markers (**C**-v).

---

**Table 1 In this example, we consider three functional groups, A, B, C.**

| Functional group | Subgroup | $G_1$ | $G_2$ | $G_3$ | $G_4$ | $G_5$ |
|---|---|---|---|---|---|---|
| A | $A_1$ | 0 | 0 | – | – | – |
|   | $A_2$ | 0 | 1 | – | – | – |
|   | $A_3$ | 1 | 0 | – | – | – |
|   | $A_4$ | 1 | 1 | – | – | – |
| B | $B_1$ | – | – | 2 | – | – |
|   | $B_2$ | – | – | 3 | – | – |
| C | $C_1$ | – | – | – | 0 | 0 |
|   | $C_2$ | – | – | – | 2 | 0 |
|   | $C_3$ | – | – | – | 0 | 2 |
|   | $C_4$ | – | – | – | 2 | 2 |

Across all groups, we consider a homoskedastic variance $\sigma I_k$, with $\sigma = 0.5$, where $I_2$ is the identity matrix and $k$ is the number of genes determining the group. Group A is determined by two genes with multivariate normal gene expression with means [0, 0], [0, 1], [1, 0], and [1, 1], group B is determined by one gene with multivariate normal gene expression means [2] (mode B1) and [3] (mode B2). Similarly, group C is determined by two genes with multivariate normal gene expression means [0, 0], [2, 0], [0, 2], and [2, 2]. Cell labels are identified through functional group tuples $(A_i, B_j, C_k)$. In this example, we have $32 = 4 \times 4 \times 2$ possible cell types. For instance, if we consider the cell types $T_1 = (A_1, B_1, C_4)$ and $T_2 = (A_2, B_2, C_3)$ it would suffice to use $G_3$ as a marker to distinguish both cell types. However, if we consider all possible 32 cell types, one needs all five genes to distinguish them. We use this functional groups example in Fig. 1C and Supplementary Figs. 3 and 4.

---

While there is no unique correct marker set, the probability of randomly selecting a correct set of $k$ markers is $(250/10,000)^k$.

**Synthetic multimodal gene expression.** We next considered a synthetic model based on the intuition that cell labels may be defined by nonadditive gene interactions. We illustrate this through simulations in which gene expression is multimodal, and synthetic cell labels are defined in combinatorial ways, such as an exclusive-or (xor) relationship among a pair of genes (Fig. 1C-i–v, Table 1, and Supplementary Figs. 2–6).

For visualization purposes, we first simulated a small dataset of synthetic gene expression profiles with three distinct functional gene groups determining ten cell types. Data were simulated from multivariate normal mixtures with means specified in Table 1 as follows. A cell label is defined by a tuple of modes $(A_i, B_j, C_k)$, where $i = 1, 2, 3, 4$, $j = 1, 2$, and $k = 1, 2, 3, 4$. The gene expression levels of each of the five markers were drawn from a Gaussian distribution, where the mean is determined by the type according to Table 1 (Fig. 1C).

We compared the performance of scGeneFit and one-vs-all in terms of true positive and false positive rates, using receiver-operating characteristic curves (Fig. 1C and Supplementary Figs. 3 and 4). While scGeneFit optimizes for joint cell label separation, it is able to outperform one-vs-all for most synthetic cell labels in one-vs-all tasks (which, by design, should be favorable to the one-vs-all marker selection methods). The performance of scGeneFit

scales favorably with the number of markers used (Supplementary Figs. 3–6). This improvement is maintained when considering a larger simulation with 15 functional groups and 40 classes across 1000 cells and 10,000 genes (Supplementary Information, Supplementary Figs. 5 and 6). In this case, data were simulated from both a multivariate normal model (Supplementary Fig. 5) and a generalized multivariate normal model with a Poisson link (Supplementary Fig. 6). In both cases, scGeneFit substantially improved classification accuracy as the number of markers increased. This means that not only does scGeneFit achieve favorable accuracy with fewer markers when theoretically possible, but scGeneFit also provides a principled approach for the selection of informative markers, when the number of markers allowed experimentally exceeds the number of classes we wish to distinguish.

**Markers for flat clustering.** We studied the performance of scGeneFit in the context of two scRNA-seq studies. To do this, we applied scGeneFit to a cord blood mononuclear cell (CBMC) study containing isolated cells from cord blood[4]. In total, 8584 single-cell expression profiles of CBMCs from the 500 most variable genes from both human and mouse cell lines were considered. For the cell type labels, we used the reported transcriptome-based clustering that partitions CBMC types into B cells, T cells, natural killer cells, monocytes (CD14+ and CD16+), dendritic cells (DC, pDC), erythrocytes, and erythroblasts. We found that the discovered marker sets were able to recover nearly identical cell type partitions in the CBMCs. We explored the space of marker set size close to the number of cell type labels, and found that this number affects the ability to discriminate among cell labels (Fig. 2A, B). We quantified the discriminatory power of the identified marker set by performing $k$-nearest neighbor clustering on the projections of cells in low-dimensional marker space (see "Methods"). We found that the distinctions between cell type labels are largely preserved in the reduced dimensional space (Fig. 2B). scGeneFit achieved slightly better performance with fewer markers than the curated marker set based on differential association (8.05% scGeneFit label classification error vs 9.90% curated marker set error in held-out data; see "Methods"), and substantially outperformed a random marker set (30% error in held-out data).

**Markers preserving hierarchical clustering.** scGeneFit has similarly good performance in the context of scRNA-seq data characterizing mouse cortical cell type diversity[14]. Among the 48 cell types, cells in the mouse somatosensory cortex are organized into a hierarchy governed by five main neuronal and neuroglial types: pyramidal neurons, interneurons, astrocytes, oligodendrocytes, and microglia. We applied the hierarchical version of scGeneFit to identify 30 marker genes that allow recovery of the hierarchical label structure (Fig. 2C, D). Our marker selection technique enabled accurate recovery of the hierarchy of cell labels. These results

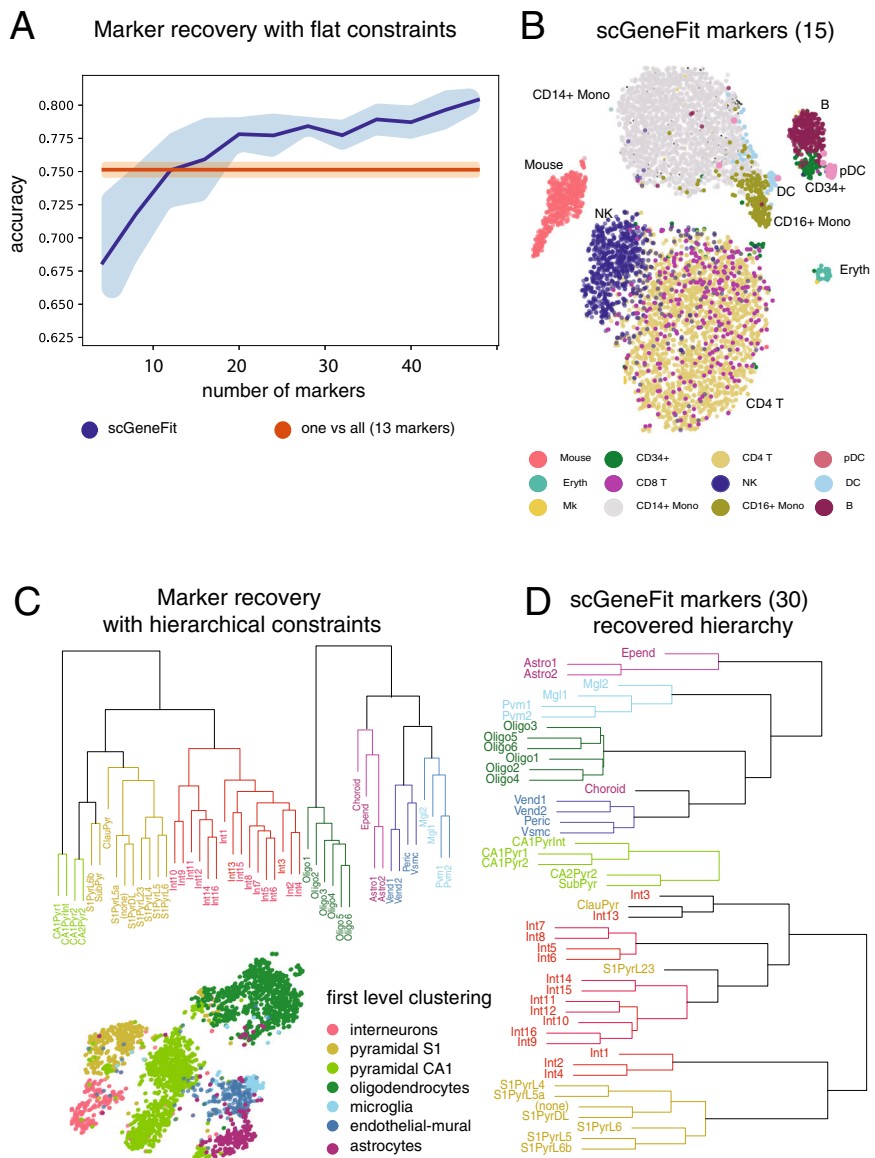

**Fig. 2 scGeneFit applied to scRNA-seq data with input cell labels both unstructured (flat) and hierarchical. A, B** Results from single-cell expression profiles of cord blood mononuclear cells (CBMC) given a flat partition of labels[4]. **A** Mean accuracy and variance of scGeneFit as a function of the number of allowed markers. **B** t-SNE visualization of scGeneFit with 15 marker genes distinguishing 13 distinct cell populations. **C** Hierarchical clustering of brain scRNA-seq data[14] and a t-SNE plot of the cell labels at the first level of the hierarchy: interneurons, pyramidal S1 cells, pyramidal CA1 cells, oligodendrocytes, microglia, endothelial–mural cells, and astrocytes. **D** Hierarchical labeling of the data with respect to the 30 markers chosen by scGeneFit. scGeneFit achieves predictive accuracy comparable to the full gene set, using 40% fewer markers than one-vs-all.

coincide with those obtained by considering >50% more markers in the one-vs-all setup (Supplementary Information, Supplementary Fig. 8). At the first level of the hierarchy, scGeneFit performed on par with the one-vs-all method, slightly improving on identification accuracy of the well understood astrocyte population (f1-score of 0.81 vs 0.74 in one-vs-all, Supplementary Information Table 4).

We compared the results obtained by scGeneFit at the second layer of the hierarchy using 30 or 40 markers with the one-vs-all method, using 48 markers (Supplementary Information). Our method with 30 and 40 markers performed on par with the one-vs-all method using 48 markers (classification error at the first layer of the hierarchy on held-out cells: 8.75% (30), 6.44% (40), 9.74% (one-vs-all); see Supplementary Table 2). In particular, scGeneFit performed well in the astrocyte, pyramidal neuron, and oligodendrocyte subpopulations (Supplementary Table 4).

## Discussion
In summary, we develop a method, scGeneFit, that identifies markers to distinguish labeled cells given a structured partition (flat partition or hierarchy) of cell labels. We show that scGeneFit is able to accurately recover a set of markers of a prespecified size and allows robust labeling of cell types in scRNA-seq data. scGeneFit is able to handle more complex relationships among class labels than a flat partition, making it the first approach to exploit hierarchical label structure to robustly solve the marker selection problem in an efficient and principled fashion. A key underlying assumption common to both scGeneFit and existing one-vs-all methods is that cell type classes are linearly separable, which is a reasonable assumption in high dimensions, but may not be true in lower dimensions. In future work, we plan to depart from this assumption by considering a more general class

of nonlinear dimension reduction methods. Furthermore, we envision relaxing the categorical labeling to a manifold constraint that will allow selection of markers to place unlabeled cells at, for example, time points along cell trajectories or locations in spatial assays.

## Methods

**scGeneFit.** In the marker selection problem, $x_i \in \mathbb{R}^d$ is gene expression measurements of the $i$th cell for $d$ different genes. We assume the subset of the cells used for training include labels.

*Setup.* We model the marker selection problem as a label-aware dimension reduction method inspired by compressive classification and largest margin nearest neighbor algorithms[13]. One such method, SqueezeFit, aims to find a projection to the lowest-dimensional subspace for which samples with different labels remain farther apart than samples with the same label. Consider a dataset $\mathcal{D} = \{(x_i, y_i)\}_{i \in \mathcal{I}}$ in $\mathbb{R}^d \times [k]$, here $x_i$ is a sample and $y_i$ is its corresponding label. We denote $\mathcal{Z}(\mathcal{D}) := \{x_i - x_j : i, j \in \mathcal{I}, y_i \neq y_j\}$ as the vector difference between samples with different labels.

The following optimization problem corresponds to finding the orthogonal projection to the lowest-dimensional space that maintains a prescribed separation $\Delta > 0$ between samples with different labels:

$$\text{minimize} \quad \text{rank } \Pi$$
$$\text{s.t.} \quad \| \Pi z \| \geq \Delta \quad \forall z \in \mathcal{Z}(\mathcal{D}), \quad \Pi^\top = \Pi, \quad \Pi^2 = \Pi. \tag{1}$$

Here, $\Pi$ is the low-dimensional projection, and $\Delta > 0$ is the desired minimum distance between projected samples $\Pi x_i$ and $\Pi x_j$ with different labels. This parameter reflects a fundamental tension in compressive classification: $\Delta$ should be large so as to enable sufficient separation of samples with different labels in the low-dimensional space, and simultaneously the projected space rank $\Pi$ should be of low dimension so that this projection effectively reduces the dimension of the sample. To address the intractability of the optimization in Eq. (1), a convex relaxation technique is used[13]:

$$\text{minimize} \quad \text{tr } M$$
$$\text{s.t.} \quad z^\top M z \geq \Delta^2 \; \forall z \in \mathcal{Z}(\mathcal{D}), \quad 0 \preceq M \preceq I. \tag{2}$$

The relaxation extends the feasible set from the set of orthogonal projections, where optimization is intractable—matrices $\Pi$ that satisfy the constraints in Eq. (1)—to the set of positive semidefinite matrices—matrices $M$ that satisfy the constraints in Eq. (2)—where one can use standard optimization toolboxes to find the global optimum in polynomial time[15]. The trace norm of $M$ corresponds to the $\ell_1$-norm of the vector of eigenvalues of $M$. Therefore, minimizing the trace norm tr M encourages $M$ to be low rank[16].

**Marker selection.** scGeneFit finds a prescribed number of gene markers, so that when the samples are projected onto those marker dimensions they exhibit the same separation of cells with different labels as in the original gene space. The objective of selecting a handful of marker genes in mathematical terms translates to finding a projection onto a subset of the coordinates; specifically, $M$ is a diagonal matrix with entries $\alpha_1, \ldots, \alpha_d$. This constraint simplifies the optimization to a linear program:

$$\text{minimize} \quad \| \alpha \|_1$$
$$\text{s.t.} \quad \sum_{j=1}^d \alpha_j z_j^2 \geq \Delta^2 \; \forall z \in \mathcal{Z}(\mathcal{D}), \quad 0 \leq \alpha_j \leq 1. \tag{3}$$

The objective's same $\ell_1$ trace norm promotes sparsity in the matrix $M$ (ref. [16]). As a result, numerical experiments show that the solution of Eq. (3) is in fact sparse, and the dimension of the projection—the number of selected markers—is smaller than the dimension of the original space.

In order for this method to be useful in practice, we modify the optimization formulation Eq. (3) to allow for outliers, and we specify the dimension of the projected space (i.e., the number of markers) $s$, leading to the scGeneFit optimization problem:

$$\text{minimize} \quad \| \beta \|_1$$
$$\text{s.t.} \quad \sum_{j=1}^d \alpha_j z_j^2 \geq \Delta^2 - \beta_z \; \forall z \in \mathcal{Z}(\mathcal{D}),$$
$$\| \alpha \|_1 \leq s, \quad 0 \leq \alpha_i \leq 1, \quad \beta_z \geq 0. \tag{4}$$

Here $\beta$ is a slack vector that quantifies how much the margin between sets with different labels is violated for each constraint[17]. $\beta$ is indexed by the elements $z \in \mathcal{Z}(\mathcal{D})$ and its dimension equals that of the constraint set $\mathcal{Z}(\mathcal{D})$.

**Incorporating label hierarchies.** Consider a hierarchical partition of the samples denoted by $T_\sigma$, where $\sigma$ is an ordered set of indices. Say $T_{\sigma'} \subset T_\sigma$, if $\sigma$ is a prefix of $\sigma'$ (for instance $T_{ijk} \subset T_{ij} \subset T_i$, corresponding to a three-level hierarchy; see Fig. 3 for a concrete example).

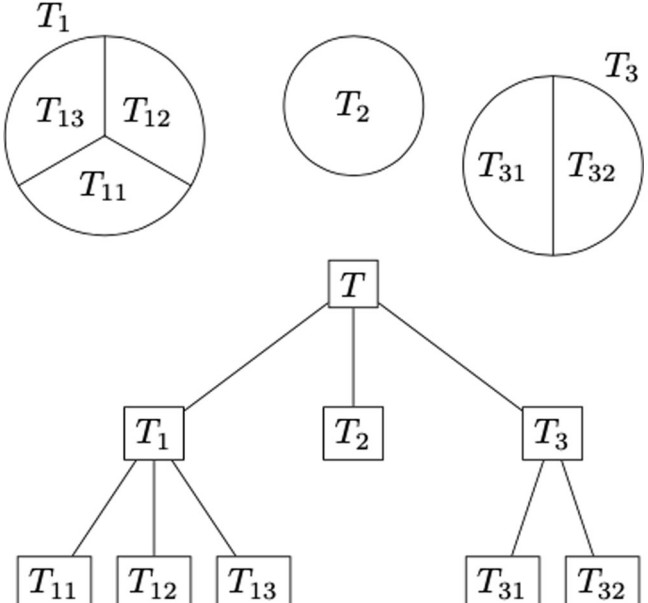

**Fig. 3 Example of hierarchical partition explaining the notation.** In this example, we have three classes ($T_1$, $T_2$, and $T_3$) at the first level of the hierarchy. At the second level of the hierarchy, $T_1$ is divided into three classes ($T_{11}$, $T_{12}$, and $T_{13}$), and $T_2$ is divided in two classes ($T_{21}$ and $T_{23}$).

When provided with the structured relationship of the labels, scGeneFit solves the optimization problem (Eq. (4)), replacing the set of constraints $\mathcal{Z}(\mathcal{D})$ to $\mathcal{Z}_T(\mathcal{D})$ to reflect the hierarchical information. In detail,

$$\mathcal{Z}_T(\mathcal{D}) := \{x_i - x_j : x_i \in T_{\sigma a}, x_j \in T_{\sigma b}, a \neq b, \text{ prefix } \sigma\} \tag{5}$$

**Alternative optimization constraints.** The optimization problem described above for scGeneFit is effective when the label structure is a flat (star shaped) hierarchy; however, when the label structure has additional layers, we would like to add an additional constraint to encourage labels that are closer in hierarchical space to also be closer in the projected (marker) space. In particular, let $\mathcal{I}_t$ be the set of indices $i \in \{1, \ldots, n\}$ of cell profiles with label $t$, and let $n_t$ be the number of cells in that set. The projected center of the profiles labeled $t$ is $\Pi c_t$, with $c_t = \frac{1}{n_t} \sum_{i \in \mathcal{I}_t} x_i$. The desired constraints can thus be formally encoded as $\| \Pi x_i - \Pi c_{t'} \|^2 - \| \Pi x_i - \Pi c_t \|^2 \geq \Delta^2$, if $y_i = t$ and $t \neq t'$.

The hierarchical scGeneFit objective encodes the intuition that the distance between labeled cells should reflect the label distance in the given label hierarchy (Supplementary Information). This is given by the linear program

$$\text{minimize} \quad \| \beta \|_1$$
$$\text{s.t.} \quad \sum_{j=1}^d \alpha_j \left[ (x_i - c_{t'})_j^2 - (x_i - c_t)_j^2 \right] \geq \Delta^2 - \beta_{i,t'}$$
$$\forall t, t' \neq t, i \in \mathcal{I}_t, \| \alpha \|_1 \leq s, 0 \leq \alpha_i \leq 1, \beta_{i,t'} \geq 0, \tag{6}$$

where, as before, $\beta$ is a slack vector.

**High-dimensional optimization.** The optimization procedure allows the analysis of thousands of genes at a time as follows. As before, let $c_t$ be the empirical gene expression mean of a class $t$ and consider constraints of the form constraints of the form $\| c_t - c_s \| > \Delta$. To insure that the number of constraints remains within the order of the number of classes, we only consider constraints over the cells with gene expression profiles closest to the cell type centroids in the lower-dimensional space. Our current implementation finds 50 markers in a simulated dataset with 10,000 cells (40 synthetic cell labels), with 10,000 genes in ~15 min (using a standard MacBook pro laptop)

**Hyperparameter setting.** In scGeneFit, the two main hyperparameters are: $s$ (the target number of markers) and $\Delta$ (the target separation of samples with different labels or centers of different classes). In our code, we implement a dual annealing method that optimize for the value of $\Delta$ for a given training set, test set, and classifier. The other hyperparameters scGeneFit uses are set to make the problem smaller in case the computational power doesn't allow the user to run the optimization in the entire dataset (like capping the number of constraints to be used or

sampling the dataset to generate fewer constraints). Such hyperparameters are fully described in the Supplementary Material.

**Optimization of the linear program and scalability**. The optimization problem (Eq. (4)) is a linear program that we solve with scipy linear programming solver (scipy.optimize.linprog). The computational bottleneck of the linear program is the number of constraints in $\mathcal{Z}(\mathcal{D})$, which a priori scales quadratically with the number of cells. In order to resolve this issue and make the optimization more efficient, we use several strategies. The simplest one is to select the most relevant constraints in Eq. (4) by considering, for each sample, the $K$-nearest neighbors from each of the other classes. Another strategy we use is to randomly select a subsample, run scGeneFit on the subsample, and project the held-out samples using the markers chosen on the subsample.

The most efficient strategy we use is to set constraints based on the empirical centroids of clusters, as discussed in the high-dimensional example above. However, such a strategy has the underlying assumption that classes are linearly separable.

A detailed comparison among all variants of our method is documented in our software release[18].

**Runtime**. The computational complexity of linear programming is an open problem in optimization, but it is known to be asymptotically upper bounded by $O(N^{2.5})$, where $N$ is the size of the problem (number of variables plus number of constraints)[19]. For the particular experiments we perform, we solve scGeneFit with 4000 variables and 6000 constraints in <40 s, on Matlab 2018a running on an Intel Xeon CPU 1.90 GHz using <4 Gb of memory.

**Dataset description and preprocessing**

*Zeisel*. Cells in the mouse somatosensory cortex (S1) and hippocampal CA1 region were classified based on 3005 single-cell transcriptomes via scRNA-seq. The nine major molecularly distinct classes of cells (layer 1) were obtained through a divisive biclustering method, and corresponding subclasses of cells (layer 2) were obtained through repeating the biclustering method within each major class[14].

*CBMC*. The CBMCs were produced with CITE-seq[4]. Single-cell RNA data processing and filtering were performed as specified in ref. [4]. In particular, the data are sparse and normalized by $\log_2(1 + X)$.

**Evaluation metrics**. In order to evaluate the performance of scGeneFit, we first split the data in training (70%) and test (30%). We train a $K$-nearest neighbor classifier on the training data (for $K = 3, 5, 15$) after projection to the corresponding markers (computed on the entire dataset). We evaluate the classifier on the test data and report the misclassification error with respect to the known classes (Supplementary Table S1). We also evaluate the performance of $k$-means clustering, using $k$-means++, reporting the smallest misclassification error among ten random initializations. For the hierarchical dataset in Zeisel, we evaluate the performance at the second level. Finally, we provide precision, recall, and $f1$-metrics for the classification tasks of both synthetic and real datasets (Supplementary Information).

**Reporting summary**. Further information on research design is available in the Nature Research Reporting Summary linked to this article.

## Data availability
The processed data we used in the experiments is available in the scGeneFit-python Github repository[18], as well as the scGeneFit package distributed by the Python Package Index.

## Code availability
We produced a python package that implements the marker selection algorithm described in this paper[18]. One can simply install the package `pip install scGeneFit`. The code to conduct the simulations and reproduce the analyses is available at https://github.com/solevillar/scGeneFit-python.

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

## Acknowledgements
This work was performed while S.V. was with New York University, and B.D. was with Princeton University. S.V. was partially supported by NSF DMS 2044349, EOARD FA9550-18-1-7007, and the NSF-Simons Research Collaboration on the Mathematical and Scientific Foundations of Deep Learning (MoDL; NSF DMS 2031985). D.G.M. was partially supported by AFOSR FA9550-18-1-0107 and NSF DMS 1829955. B.E.E. was supported by a grant from the Helmsley Trust, a grant from the NIH Human Tumor Atlas Research Program, NIH NHLBI R01 HL133218, and NSF CAREER AWD1005627.

## Author contributions
B.D. and S.V. conceived the idea, designed and implemented the method, and ran the experiments. S.V., B.D., and B.E.E. analyzed the results and drafted the manuscript. D.M. provided guidance regarding the optimization framework for dimensionality reduction. All authors read and approved the final manuscript.

## Competing interests
B.E.E. is on the Scientific Advisory Board of Freenome, Celsius Therapeutics, and Crayon Bio. B.E.E. is a consultant for Freenome and was employed by Genomics plc during 2019-2020. Otherwise, the authors declare that they have no additional competing interests.
