## [Peer Review File · Nature Communications]

Reviewers' comments:

Reviewer #1 (Remarks to the Author):

In this manuscript, Dumitrascu et al. address the problem of marker selection for single-cell RNA-seq data, whereby a set of cells sequenced using scRNA-seq are labelled (either through physical sorting or in-silico annotation) and we wish to select a set of genes ("markers") that best represent the cell for further downstream selection / interpretation. The authors introduce scGeneFit, a model based on a previous label aware dimension reduction method, but in which rather than each reduced dimension constituting a linear combination of features (genes), it constitutes the selection of a single gene (marker) with only a set number having non zero weight. The authors apply their method to some simulated data and two datasets of real scRNAseq.

Overall, the problem is timely in the field and the proposed model is reasonable. However, the analysis presented is severely under-developed to be convincing of the effectiveness of scGeneFit. In particular, (i) the simulations are unrealistic and overly restrictive, (ii) no simpler methods are compared against to convince of the need for scGeneFit, (iii) the evaluation criteria on real data needs improved, and (iv) too little detail is presented in the paper to faithfully reproduce the results. I have detailed major comments and suggestions below that I think if addressed would greatly improve the paper.

Major

Comparison to existing / simpler approaches. The authors cite two existing approaches for marker identification, and there are others used in the single cell field (e.g. the findMarkers() function in the scran package). Further, a recent method [1] to select "informative" genes could be applied to each cluster to derive a set of marker genes, and one could derive an even simpler approach by taking just the top s most variable genes. The authors should attempt to compare against these simpler methods in all simulations and real data comparisons, to get a sense of the gain from scGeneFit.

[1] <https://journals.plos.org/plosone/article?id=10.1371/journal.pone.0210571>

Simulations: the current simulation strategy is not comprehensive enough to evaluate scGeneFit. The authors should do the following:

- Simulate from a suitable noise model (e.g. negative binomial or multinomial using splatter) rather than from Gaussian distributions
- Only a single number of genes (200) with 50 as markers for 3 cell types is considered. 200 is unrealistically small starting number of genes for scRNAseq and this simulation implies that $150/200 = 75\%$ of genes are markers, which makes finding markers relatively easy. The authors should benchmark with a constant set of overall genes (e.g. 10000 is roughly standard for the # expressed in scRNA-seq, though some pre filtering may change this, which should be taken into consideration) as the number of markers increases.
- If we know the ground truth genes through simulations, what are the false positive and false negative rates of finding these? The evaluation in 1B seems odd given it doesn't show that markers are recovered for a given cell type (is this found post-hoc?) and an equivalently good set of markers could be recovered by taking the top s most variable genes.
- How does scGeneFit perform as the number of specified markers (s) changes with respect to the number of true markers? In the simulation the authors set $s = 20$, but how does this vary as s is changed? How should s be determined in general? Is there a way to infer s or any recommendations for end users?
- Mutual exclusivity of markers: for the simulation considered, 50 markers are used for 3 cell types = 150 total, which means these are mutually exclusive, but there is no reason to expect this to be the case, ie we can have a marker (e.g. CD45) that is a marker for multiple cell types. How does the performance of scGeneFit change as the overlap between these sets increases?
- Correlated noise: the current simulation setup has uncorrelated noise for non marker genes, but in general this is unrealistic as other regulatory pathways such as cell cycle stage will induce

correlations - what happens if this is taken into account in the simulation from work, ie
Expression | cell type, cell cycle stage ~ coefficient * cell cycle stage + over-expression for cell
type if marker

Evaluation: the authors should evaluate the simulations in terms of precision, recall, etc, since it's essentially a classification problem (classifying genes as markers or not). For the real data, evaluation appears to be variably performed by eye and then by clustering on the marker gene only data using knn on a test set. Can the authors show this is robust to more than one clustering algorithm (e.g. use Seurat and SC3 also), and report e.g. precision recall curves for all cell types rather than just accuracy.

On the CMBC dataset the authors apply scGeneFit across a range of s values, and report differing accuracy. What does this function look like across a wide range of s? Does it have a maximum value (that would give a procedure for picking s) or does it simply increase as a function of s?

A "curated marker set" is referred to in main text but doesn't appear in methods?

Software: given scGeneFit should be a tool for the community to use to maximize impact, the current implementation on github is written in a proprietary language (matlab) with no documentation or examples will seriously preclude adoption. Though this doesn't on its own invalidate the paper, it greatly reduces the utility of scGeneFit as a tool.

Hyper parameter settings: the method has two free hyperparameters. A section in the methods implies to help with hyperparameter tuning, but does not actually help ("A rule of thumb is to choose Δ as a function of the separation between the classes" does not actually provide advice). The authors should (i) detail the values of Δ used in the paper, (ii) show the performance of scGeneFit is robust to Δ in simulations + real data, (iii) include practical guidance on how to set Δ (and s as above).

Minor

The paper would benefit from section headings to improve readability

The authors refer to "reduced dimensionality space" several times, which on a first read I assumed to mean the tSNE space rather than the marker gene space. Though technically correct, it's worth changing to "marker gene space" or similar to avoid confusion

Reviewer #2 (Remarks to the Author):

The authors present a novel computation tool, scGeneFit, which identifies subsets of marker genes of a specified size that are able to distinguish cells belonging to different cell-types. They this method demonstrate on simulated data and two real single-cell datasets. This is potentially a very useful tool to aid the validation and guide follow-up experiments from single-cell RNAseq results.

Major Concerns:

1. It is unclear whether scGeneFit is a significant improvement over the existing one-vs-all approach. The authors only systematically compare these two approaches on two real single-cell datasets, the CBMC and Zeisel, for the former scGeneFit performs about the same as one-vs-all (~5% improvement) when using the same number of markers and for the latter scGeneFit performs clearly worse (10-30% decrease) than one-vs-all when using the same number of markers. Nor do the authors show that scGeneFit is significantly faster nor capable of selecting smaller marker sets than the one-vs-all approach.
2. In addition, the authors do not compare their method to the main competing method: COMET (<https://www.biorxiv.org/content/10.1101/655753v1>)
3. The tool is currently written in MATLAB which requires a licence to use, thus I was unable to test the tool myself. Re-implementing the tool in an open-source language such as R or Python and making it compatible with at least one existing single-cell RNA-seq framework: e.g.

SingleCellExperiment, Seurat, or scanpy, would make it much more usable by the community.

4. Many of the results are presented qualitatively using t-SNE plots. Multiple t-SNEs are difficult to compare since the projected coordinates are sensitive to the number of cells, perplexity parameters, and stochastic effects of the algorithm. In addition, dense t-SNE plots may have important results obscured by overlapping points. The manuscript would be much improved by presenting the results in a more quantitative manner (such as Table 1 of the supplementary), to make the comparisons between different methods and different numbers of markers on the benchmarking datasets more clear.

5. scGeneFit requires users to specify the number of genes to select without providing any guidance on how to select this parameter. In addition, in many instances the user would be interested in the minimal number of markers necessary to achieve a particular accuracy/precision in cell-classification, which is not directly answerable with scGeneFit currently. Could the author provide an option to use the method to identify the minimal markers to achieve a given accuracy in cell-type classification?

6. scGeneFit will be of most use for designing follow-up FACS gating strategies or FISH-based imaging experiments. However, these approaches are constrained on which genes they use, the former to surface proteins the latter to transcripts with low homology to other transcripts. Is it possible to constrain scGeneFit to only consider genes compatible with the particular downstream experiment one is interested in?

Minor Concerns:

7. I'm concerned the authors failed to differentiate microglia from other cell-types in their demonstration dataset. In our experience microglia are very clearly distinct from other brain cell-types with many robust markers due to their unique immunological properties. Perhaps if the authors did not limit themselves to the highly-variable genes of the dataset these markers would be identified by their tool.

8. The colour scheme chosen for Figure 1B makes it hard to distinguish the four cell-clusters in panel iii. It's also not clear what the scale is for the expression values for the heatmaps in this figure.

9. By what measure do the authors consider the hierarchies in Figure 2 A&B as very similar to each other? Much of the topology is different between these trees both at a coarse grained level - e.g. the relationship of oligodendrocytes to other glial cell-types, and fine grained level - e.g.

Response to Reviews

We thank the reviewers for their thoughtful comments. We first acknowledge that a long time has passed since we received the reviews for this paper. We would like to point out that during this time we have engaged with the larger computational biology community and incorporated feedback from collaborators and researchers across the world who have started using our software in their projects. Our original code was developed in MATLAB, which is proprietary software. We since have developed a *pip installable python* package. We believe this software package improves the contribution and impact of this work significantly, and it justifies the long period of time in revision. The package is available as open source and can be installed as

```
pip install scGeneFit
```

the documentation is available in <https://pypi.org/project/scGeneFit/1.0.1/>. We also produced a set of examples and jupyter notebooks that explain how to use the package. For more information see the Github repository <https://github.com/solevillar/scGeneFit-python>.

Below, we respond to each of the Reviewers' comments one-by-one. The Reviewers' comments are in blue italic text; our response is in black text, and additions to the manuscript are in red text.

Reviewer 1

Overall, the problem is timely in the field and the proposed model is reasonable. However, the analysis presented is severely under-developed to be convincing of the effectiveness of scGeneFit. In particular, (i) the simulations are unrealistic and overly restrictive, (ii) no simpler methods are compared against to convince of the need for scGeneFit, (iii) the evaluation criteria on real data needs improved, and (iv) too little detail is presented in the paper to faithfully reproduce the results.

We thank the reviewer for their comments and suggestions. Briefly, we addressed them as follows: (i) we provide a comprehensive, new software package with a new set of simulations that identify settings in which the one-vs-all methods fail (ii) we thoroughly discuss the limitations of current methods, (iii) where appropriate, we expand on our evaluation criteria, (iv) we provide additional Python code and notebooks reproducing our results, and we revisit the description of our method. The details of these major changes are discussed below.

Major Comments

Comparison

Comparison to existing / simpler approaches. The authors cite two existing approaches for marker identification, and there are others used in the single cell field (e.g. the findMarkers() function in the scan package). Further, a recent method [1] to select "informative" genes could be applied to each cluster to derive a set of marker genes, and one could derive an even simpler approach by taking just the top s most variable genes. The authors should attempt to compare against these simpler methods in all simulations and real data comparisons, to get a sense of the gain from scGeneFit.

[1] <https://journals.plos.org/plosone/article?id=10.1371/journal.pone.0210571>

In our new experiments, we compare scGeneFit with one-vs-all methods on real data on an expanded set of simulations. Our one-vs-all method follows the `findMarkers()` method in the benchmark `Seurat` package. While the default testing regime of `Seurat` is different from that of the `scrn` package that the reviewer mentions, (`Seurat` compares the expression levels in one cell label class to those of all cells outside of the cell label), we point out that both methods conceptually perform the same task: they rank genes *independently* based on how well they separate or identify one particular cell label, while disregarding gene correlation or labeled structure. In contrast, we sought to address a problem for which a solution does not yet exist: how to detect **combinatorial** gene panels that distinguish among structured cell labels when the labels are not simply obtained by clustering the data.

We thank the reviewer for pointing out to us the excellent paper of McCurdy et al. We would like to emphasize that this method does not aim to address our question. We operate in a **supervised** setting in which the cell type labels are **given** to us potentially from another modality/property. Were the cell-type labels extracted from the clustering of the gene expression data, than the McCurdy et al. paper would be relevant. Furthermore, we emphasize that there is currently **no other method available** in the computational biology community to address this **supervised** question. As the reviewer mentions, we did compare to simpler methods in the context of our “one-vs-all comparison” (we include the one-vs-all function in the package). However, outperforming “one-vs-all” in all its variations is not the goal of our paper. By providing an efficient optimization scheme to a combinatorial problem, we address two gaps in the current gene marker selection literature:

- How do we take advantage of cell label structure to select markers when the number of labels we aim to distinguish is much larger than that of experimentally allowed markers?
- How do we select markers when cell labels are identified jointly by groups of genes, rather than uniquely specified by any single gene?

Both questions arise when aiming to find gene sets that identify many rare cell types jointly in a spatial context [1] or that maintain hierarchical organization as observed in development [2, 3]. In this work, we show this is possible through means other than one-vs-all testing, in a manner that achieves similar or higher accuracy, with possibly fewer markers and circumventing expensive combinatorial search.

Simulations

Simulations: the current simulation strategy is not comprehensive enough to evaluate scGeneFit.

Our manuscript considered a simulation in which the cell labels were determined by gene covariance differences as opposed to differences in individual gene expression means. From our point of view, this is the simplest example in which one can immediately see the shortcoming of one-vs-all approaches: there is no gene that is significantly associated with the covariance-mediated labeling, therefore groups of genes can not be selected using that criterion (one-vs-all significance test). We extended our simulation strategy as follows.

Functional Group Simulation. We include a new set of biologically inspired simulations based on the intuition that genes belong to different functional groups that are used combinatorially to determine cell type. The complete description of these simulations can be found at the bottom of this document, in the Section **Additional numerical results**, which we include in the Methods and Supplementary Information sections of the manuscript. In this setting, we further evaluated the performance of `scGeneFit` with respect to noise distributed according to both Gaussian and Poisson link functions. The analysis from these additional simulations further showed the advantage of `scGeneFit` in comparison to the one-vs-all approach. We modified Figure 1 to illustrate these results, and the corresponding textual changes have been made to the manuscript.

Simulate from a suitable noise model (e.g. negative binomial or multinomial using splatter) rather than from Gaussian distributions.

Recent work on FACS sorting [4] explicitly relates the difficulties on separating rare cell types that are not linearly associated with any given channel (marker), but which can emerge through a combination of channels. Splatter is not equipped to generate such data. Splatter uses a Poisson-based simulation model that can incorporate dropout, library size, dispersion, cell label specific means, and has been successful in providing simulation benchmarks for nonnegative matrix factorization tasks [5]. However, gene expression is fed as a parameter to a cell label gene expression-specific distribution that follows the one marker-one cell type assumption and does not allow for combinatorial/pathway specific marker mediated cell label gene expression. In other words, Splatter simulations are built on the assumption that expression within cells of each cell label is based on the differential expression of individual single genes, assumption that we depart from.

Instead, we simulate our data according to a hierarchical Poisson multivariate normal model as described in Section **Additional numerical results**. This model is similar to using a Poisson Link in a generalized linear model [6]. In this setting as well, scGeneFit outperforms the one-vs-all approach: it achieves the same classification accuracy with less markers, and improves on the accuracy as the number of markers allowed increases.

-Only a single number of genes (200) with 50 as markers for 3 cell types is considered. 200 is unrealistically small starting number of genes for scRNAseq and this simulation implies that $150/200 = 75\%$ of genes are markers, which makes finding markers relatively easy. The authors should benchmark with a constant set of overall genes (e.g. 10,000 is roughly standard for the # expressed in scRNA-seq, though some pre filtering may change this, which should be taken into consideration) as the number of markers increases.

We revisited our simulations and extended our simulations and methods to allow working with larger datasets. Our current implementation finds 50 markers in a simulated dataset with 10,000 cells (40 synthetic cell types) with 10,000 genes in around 15 minutes (using a standard MacBook pro laptop). We describe our implementation choice in the Methods section through the addition of the following paragraph:

Our optimization procedure allows the analysis of thousands of genes at a time as follows. As before, let c_t be the empirical gene expression mean of a class t , and consider constraints of the form $\|c_t - c_s\| > \Delta$. To insure that the number of constraints remains within the order of the number of unique cell labels, we only consider constraints over the cells with gene expression profiles closest to the cell type centroids in the lower dimensional space.

Additional implementation details are documented in our released code. Our current implementation is quite efficient but relies on the underlying assumption that the classes are linearly separable, which is a reasonable assumption in high dimensions but may not be in lower dimensions. We include this comment in our discussion to motivate future work, and make the appropriate edits to the Methods section. Finally, in Figure 4 of this review (Supplementary Figure 4,5,6), we show the performance of scGeneFit in a larger scale problem with 10,000 cells and 10,000 genes. Each run of the scGeneFit took around 10 minutes in a Google Colab notebook (standard account).

- If we know the ground truth genes through simulations, what are the false positive and false negative rates of finding these? The evaluation in 1B seems odd given it doesn't show that markers are recovered for a given cell type (is this found post-hoc?) and an equivalently good set of markers could be recovered by taking the top s most variable genes.

We agree and we have addressed this concern. Kindly refer to our comment on providing new Evaluation Criteria as well as the results from our newest simulations (Figure 1, panel C ii)), for which we provide explicit precision recall curves. For the simulation presented in our manuscript, the markers do not identify any given cell label separately, but all the cell labels jointly. Furthermore, while the model considered is heteroskedastic, the partitioning of the covariance is what leads to a particular cell labeling, and not the highest variance from individual genes; hence the top s most variable genes do not lead to a correct solution.

How does scGeneFit perform as the number of specified markers (s) changes with respect to the number of true markers? In the simulation the authors set $s = 20$, but how does this vary as s is changed? How should s be determined in general? Is there a way to infer s or any recommendations for end users

We thank the reviewer for this suggestion. We added these results in the context of our scaled up simulations. Please find these results in our new Figure 1 (synthetic data) and Figure 2 (scRNA-seq data). In both simulated and real data **scGeneFit** outperforms one-vs-all by either achieving better performance with fewer markers or by identifying situations when more markers are needed to better distinguish among the current classes.

In detail, we plot accuracy as a function of s , the number of selected markers, which shows the accuracy achieving a plateau as we reach the number of true markers. Determining the ideal number of markers could be done using a variant of the elbow method for determining the number k of clusters in k nearest neighbors. However, we operate under the assumption in which the bottleneck is given by the maximum number of markers one can tag in an experiment: the number of markers should be chosen according to resources and desired accuracy.

Mutual exclusivity of markers: for the simulation considered, 50 markers are used for 3 cell types = 150 total, which means these are mutually exclusive, but there is no reason to expect this to be the case, ie we can have a marker (e.g. CD45) that is a marker for multiple cell types. How does the performance of scGeneFit change as the overlap between these sets increases?

Our new simulations were developed with this comment in mind. In the functional group simulation, genes are partitioned into functional groups. The functional groups can be active at different levels and thus determine cell types, with the expression of some functional groups belonging to multiple cell types. Due to its combinatorial nature, **scGeneFit** particularly shines in this setting (Figures 2,3,4 of the revised manuscript). These Figures have been added to the Supplementary Information.

*- Correlated noise: the current simulation setup has uncorrelated noise for non marker genes, but in general this is unrealistic as other regulatory pathways such as cell cycle stage will induce correlations- what happens if this is taken into account in the simulation from work, ie Expression — cell type, cell cyclestage coefficient * cell cycle stage + over-expression for cell type if marker*

While we agree with the reviewer’s comment, we emphasize that the particular way of simulating data the reviewer suggested underlies the assumption that cell label-specific gene expression is determined by the additive contribution of marker genes. This is different from the problem we are considering here (where markers are combinatorially contributing to identifying cell label). We hope our new set of simulations further addresses this point, since the noise of some non-marker genes is correlated with noise of the marker genes.

Evaluation

Evaluation: the authors should evaluate the simulations in terms of precision, recall, etc, since it’s essentially a classification problem (classifying genes as markers or not). For the real data, evaluation appears to be variably performed by eye and then by clustering on the marker gene only data using knn on a test set. Can the authors show this is robust to more than one clustering algorithm (e.g. use Seurat and SC3 also), and report e.g. precision recall curves for all cell types rather than just accuracy.

We provide precision recall-curves for all our simulations, and we add these results to our augmented Figure 1 and Supplementary Information. In particular, for each simulation, we ran *scGeneFit* with different numbers of markers, and we plotted the classification accuracy of a K-nearest neighbors classifiers as a function of the number of markers. To enable this analysis, we had to look at classification within each particular label class. **scGeneFit** outperforms the one-vs-all method across the board.

On the CMBC dataset the authors apply scGeneFit across a range of s values, and report differing accuracy. What does this function look like across a wide range of s ? Does it have a maximum value (that would give a procedure for picking s) or does it simply increase as a function of s ? We thank the reviewer for this

suggestion. We provide this plot in Panel A of the augmented Figure 2 of the main manuscript. We have added similar plots for our extended simulations as well (Supplementary Information).

A “curated marker set” is referred to in main text but doesn’t appear in methods?

We apologize for this misnomer. The marker set refers to the set obtained using the one-vs-all method. We clarified this in the text.

Proprietary Software and Replication

Software: given scGeneFit should be a tool for the community to use to maximize impact, the current implementation on github is written in a proprietary language (matlab) with no documentation or examples will seriously preclude adoption. Though this doesn’t on its own invalidate the paper, it greatly reduces the utility of scGeneFit as a tool.

We also take the existence of reproducible tools seriously. As a result, with the feedback of an extended community of researchers, we developed a python package `scGeneFit`, together with Jupyter notebooks meant to replicate our full analysis. These can be found at our github repo <https://github.com/solevillar/scGeneFit-python>.

Hyper parameter settings: the method has two free hyperparameters. A section in the methods implies to help with hyperparameter tuning, but does not actually help (“A rule of thumb is to choose Δ as a function of the separation between the classes” does not actually provide advice). The authors should (i) detail the values of Δ used in the paper, (ii) show the performance of scGeneFit is robust to Δ in simulations + real data, (iii) include practical guidance on how to set Δ (and s as above).

This is an important point. Please find a updated thorough description of the parameters and the search for their best configurations in the section **Parameter optimization** below. This section was added to the Supplementary Information to the main manuscript.

Minor Comments

The paper would benefit from section headings to improve readability.

We were attempting to follow the brief communications online guideline: The main text is typically 1,000-1,500 words, including the abstract and contains no headings with the exception of a single heading for Methods to point readers to the online Methods section providing all technical details necessary for the independent reproduction of the methodology. However, we have included section headings for clarity.

The authors refer to “reduced dimensionality space” several times, which on a first read I assumed to mean the tSNE space rather than the marker gene space. Though technically correct, it’s worth changing to “marker gene space” or similar to avoid confusion

We have also edited our text to contain “marker gene space” instead of “reduced dimensionality space” to avoid confusion.

Reviewer 2

We thank the reviewer for their thoughtful comments, which we address point by point.

Major Comments

It is unclear whether scGeneFit is a significant improvement over the existing one-vs-all approach. The authors only systematically compare these two approaches on two real single-cell datasets, the CBMC and Zeisel, for the former scGeneFit performs about the same as one-vs-all (5% improvement) when using the same number of markers and for the latter scGeneFit performs clearly worse (10-30% decrease) than one-vs-all when using the same number of markers. Nor do the authors show that scGeneFit is significantly faster nor capable of selecting smaller marker sets than the one-vs-all approach.

We consider scGeneFit to present a conceptual advantage compared to the one-vs-all approach because it provides the flexibility of prescribing the number of markers. In particular, in the hierarchical setting of the Zeisel dataset, we observe that scGeneFit achieves good performance with fewer markers than the total number of classes, which is unfeasible for one-vs-all. We do believe the issue the reviewer is pointing out is important. To this end we improved our implementation of our marker selection method, and in particular we implemented a search in the space of parameters that finds the best parameters for a dataset. Using this optimization, for CBMC we find that scGeneFit obtains comparable performance to one-vs-all with the same number of markers (77% accuracy for scGeneFit and 73% for one-vs-all). With 20 markers, scGeneFit improves its accuracy to 80%, which perhaps is a statement of how well one-vs-all performs in this particular dataset. However we argue that scGeneFit is theoretically justified and flexible enough to be considered a valuable contribution to the field in conceptual terms and as software.

In the Section **Additional Numerical Results** we show an additional set of experiments that we hope convince this reviewer of the value of scGeneFit for marker selection. In particular we show the accuracy of scGeneFit improving with the number of markers.

In addition, the authors do not compare their method to the main competing method: COMET (<https://www.biorxiv.org/content/10.1101/655753v1>)

Before we point out our main differences with COMET, we would like to stress the fact that our manuscript was in review for a long time, following its initial submission on biorxiv on April 4th 2019, while COMET appeared online, on biorxiv as well, almost **two months later**, on May 30th.

Most importantly, COMET is a one-vs-all method that provides an alternative to the Kolmogorov-Smirnov test via the XL-mHG test. While COMET claims to perform combinatorial selection, the methods reveal that it exhaustively select sets of k markers and performs a one-vs-all test on the resulting configurations. This has complexity G^k where G is the number of genes. Thus, the method becomes unfeasible as k reaches numbers as low as 4. We downloaded the package and found that we could not run it on most of our simulations and scRNA-seq data sets. Hence, while COMET is a nicely packaged XL-mHG test for gene enrichment, we do not think it properly addresses the combinatorial search problem, making it unfeasible for comparison to our test case. We now cite and include COMET in our Related Work section in the updated version of our manuscript.

The tool is currently written in MATLAB which requires a licence to use, thus I was unable to test the tool myself. Re-implementing the tool in an open-source language such as R or Python and making it compatible with at least one existing single-cell RNA-seq framework: e.g. SingleCellExperiment, Seurat, or scanpy, would make it much more usable by the community.

As mentioned before we now have an open source Python package (one can just `pip install scGeneFit`). We also have jupyter notebooks illustrating our approach at our github repository <https://github.com/solevillar/scGeneFit-python>. Here we also provide extensive simulations addressing comments from Reviewer 1 that illustrate a varied set of contexts in which `scGeneFit` shines.

Many of the results are presented qualitatively using tSNE plots. Multiple tSNEs are difficult to compare since the projected coordinates are sensitive to the number of cells, perplexity parameters, and stochastic effects of the algorithm. In addition, dense tSNE plots may have important results obscured by overlapping points. The manuscript would be much improved by presenting the results in a more quantitative manner (such as Table 1 of the supplementary), to make the comparisons between different methods and different numbers of markers on the benchmarking datasets more clear.

This is an important point, and we share this concern. We stress that we only use tSNE for the purpose of visualization, and not validation, as we are aware of its shortcomings. The metrics used for comparison are not based on tSNE, but through the classification performance of simple classifiers such as nearest centroid or k-nearest neighbors. We provide additional metrics in the shape of precision recall curves, and metrics that allow us to quantify the performance of our method in terms of both overall accuracy and per-cell label accuracy. We include these results in our augmented Figures and Tables in the main manuscript and in the

Supplementary Information.

scGeneFit requires users to specify the number of genes to select without providing any guidance on how to select this parameter. In addition, in many instances the user would be interested in the minimal number of markers necessary to achieve a particular accuracy/precision in cell-classification, which is not directly answerable with scGeneFit currently. Could the author provide an option to use the method to identify the minimal markers to achieve a given accuracy in cell-type classification?

This is an excellent question. Our original motivation for the marker selection problem considered the number of markers to be limited by a physical constraint (like in smFish). However, our software allows us to observe how the classification accuracy increases with the number of markers (see plots in Section Additional numerical results). One method to select the number of markers is through the elbow rule, which is a popular heuristic designed to find the number of clusters in a dataset in k-nearest neighbours applications.

scGeneFit will be of most use for designing follow-up FACS gating strategies or FISH-based imaging experiments. However, these approaches are constrained on which genes they use, the former to surface proteins the latter to transcripts with low homology to other transcripts. Is it possible to constrain scGeneFit to only consider genes compatible with the particular downstream experiment one is interested in?

Given a known set of genes compatible with a certain downstream experiment (e.g. surface proteins) it is possible to ask scGeneFit to select the markers from that particular subset. This can be done simply by restricting the input data. We have added this important point to our Discussion.

Minor comments

I'm concerned the authors failed to differentiate microglia from other cell-types in their demonstration dataset. In our experience microglia are very clearly distinct from other brain cell-types with many robust markers due to their unique immunological properties. Perhaps if the authors did not limit themselves to the highly-variable genes of the dataset these markers would be identified by their tool.

We expanded our analysis to include precision-recall curves and per-cell label accuracy. Our analysis shows that microglia are well-differentiated in the dataset. In particular, when considering the downstream task of classifying microglia, the markers detected by scGeneFit resulted in 0.87 precision and 0.90 recall, while the markers from the one-vs-all method resulted in a substantially worse 0.70 precision and 0.74 recall. We include the precision-recall results for microglia as well as those for the other classes of the first level of the hierarchy in Supplementary Information. The statement in the paper regarding microglia referred to the lower level of the hierarchy, where some microglia subtypes are more difficult to distinguished from one another. The statement was not referring to the class of microglia. We removed it to avoid confusion.

The colour scheme chosen for Figure 1B makes it hard to distinguish the four cell-clusters in panel iii. It's also not clear what the scale is for the expression values for the heatmaps in this figure.

We apologize for this inconvenience - we changed the plot to a color blind palette which we hope is more easily distinguishable. We also included legends for all our figures, including those containing heatmaps. To incorporate the feedback from both reviewers, we modified the captions and content of the figures of our main manuscript. We now have two thematically distinct figures: Figure 1 (main manuscript) illustrates the method and visualizes results from synthetic datasets and Figure 2 (main manuscript) visualizes results obtained on real data with unstructured cell labels and in hierarchically structured cell labels. The first figure represents synthetic data that is described in the main manuscript.

By what measure do the authors consider the hierarchies in Figure 2 A&B as very similar to each other? Much of the topology is different between these trees both at a coarse grained level - e.g. the relationship of oligodendrocytes to other glial cell-types, and fine grained level - e.g.

The hierarchies provided in Figure 2 are for visualization only.

While there exist ways to measure distances between trees (e.g., the Robinson-Foulds metric is used for phylogenetic trees), analyzing such metrics is out of the scope of this paper. Instead, for the second level

of the hierarchy (48 cell type labels), we evaluate the recovery rate of a nearest centroid cell label classifier. When trained on the entire dataset (i.e., containing all genes as markers), the classification accuracy at the second level of the hierarchy is a meager 0.62, which explains the variability in hierarchy visualization. When restricted to 48 markers obtained by scGeneFit and one-vs all, the classifier has recovery rates of 0.64 and 0.61 respectively. However, scGeneFit can obtain a similar recovery rate with fewer markers, for instance, when restricted to 30 markers, the recovery rate at the second level of the hierarchy is 0.60. This shows that redundancy in the data can be exploited to allow data recovery with fewer markers than one-vs-all methodology.

Furthermore, we provide precision and recall values from the classification task for the first level of the hierarchy. We refer to these results in the main manuscript, and provide a full summary of this analysis in Table 3 of the Supplementary Information. As our knowledge of cell types expands and diversifies, especially in the context of the Human Cell Atlas or Single Cell eQTL Atlas, we hope scGeneFit will fill an important methodological conceptual gap, as well as stimulate the interest of the broader optimization and machine learning community to contribute to biological problems.

Additional numerical results

In this section, we consider a set of simulations in which we compare and contrast `scGeneFit` and one-vs-all marker selection procedures.

Synthetic functional group model

We consider a synthetic model based on the intuition that genes interact with one another in a complex manner (e.g their products belong to related pathways) giving rise to biological modules or functional groups which are utilized differently across cell types. Here, we consider functional groups to be determined by a (possibly overlapping) set of genes. Every functional group has a multi-modal structure: groups of genes can have different levels of activity (subgroups). Cell types are then determined by a unique combination of the different instances of the functional groups (see Table 1 and Figures 1, 2 for a specific example). To allow for clear visualization and metric evaluation, we consider a small toy example with 9 cell types and only 100 cells. The small toy example makes it easier to observe the effect the marker selection at the individual label level. In particular, we compare and contrast the performance of the two approaches in terms of true positive and false positive rates, as illustrated by Receiver-operating characteristic curves. This shows that while scGeneFit optimizes for joint separation, it is able to outperform one-vs-all for most synthetic cell types in one-vs-all tasks (which, by design, should be favorable to the one-vs-all marker selection procedures).

Single cell count data

In Figure 3, as suggested by the reviewers, we consider a count based model. The data is generated using a Poisson link function $\log(\text{Poisson}(\exp(X_{ij})) + 1)$, element wise, where X_{ij} is an element of the matrix X , generated from the mixture of multivariate normal distributions defined by the functional group example (Figures 1 and 2). This follows statistical models with a rich history in describing count data [7, 8]. In Figure 5, a similar model is explored in a real dataset scale setting.

Large scale computation

In Figure 5 we consider a larger scale version of the example we described above. In this larger scale experiment, we generate 40 classes from 15 different functional groups. Each gene is randomly assigned to a gene expression subgroup. While in this case there are multiple marker sets that are suitable for discriminating among a given cell labeling, the probability of detecting a good marker set is negligible ($O(1/(40)^{15})$).

Functional group	subgroup	G_1	G_2	G_3	G_4	G_5
A	A_1	0	0	-	-	-
	A_2	0	1	-	-	-
	A_3	1	0	-	-	-
	A_4	1	1	-	-	-
B	B_1	-	-	2	-	-
	B_2	-	-	3	-	-
C	C_1	-	-	-	0	0
	C_2	-	-	-	2	0
	C_3	-	-	-	0	2
	C_4	-	-	-	2	2

Table 1: In this example we consider 3 functional groups, A, B, C . Across all groups we consider a homoskedastic variance σI_k with $\sigma = 0.5$ where I_2 is the identity matrix and k is the number of genes determining the group. Group A is determined by two genes with multivariate normal gene expression with means $[0,0]$, $[0,1]$, $[1,0]$, and $[1,1]$, group B is determined by one gene with multivariate normal gene expression means $[2]$ (mode B_1) and $[3]$ (mode B_2). Similarly, group C is determined by two genes with multivariate normal gene expression means $[0,0]$, $[2,0]$, $[0,2]$, and $[2,2]$. The cell types are identified through functional group tuples (A_i, B_j, C_k) . In this example we have $32 = 4 * 4 * 2$ possible cell types. For instance, if we consider the cell types $T_1 = (A_1, B_1, C_4)$ and $T_2 = (A_2, B_2, C_3)$ it would suffice to use G_3 as a marker to distinguish both cell types. However, if we consider all possible 32 cell types one needs all 5 genes to distinguish them. We use this functional groups example in Figures 1 and 2

In this specific example with 1000 cells and 10,000 genes, `scGeneFit` finds the number of markers in 10-12 minutes in a standard MacBook Pro (or a Google Colab on a standard free account). In order to make our algorithm work in this larger scale setting we modify our method as explained in the next Section as well as in the methods Section of our manuscript.

In Figure 5 we show the performance of `scGeneFit` in a larger scale problem and its comparison with one-vs-all.

Parameter optimization

Our main method `get_markers` receives the following set of parameters:

- **data:** $N \times d$ numpy array with cells genetic expression, N : number of cells (i.e. “points”), d : number of genes (i.e. “dimension”).
- **labels:** list with labels (N labels, one per point)
- **num_markers:** target number of markers to select (`num_markers < d`).
- **method:** ‘centers’, ‘pairwise’, or ‘pairwise_centers’
 - ‘centers’ considers constraints that require that two consecutive classes have their empirical centers separated after projection to the selected markers. According to our numerical experiments this is the least general but most efficient and stable set of constraints (the underlying assumption is that the classes are linearly separable, which seems to be true in high dimensions).
 - ‘pairwise’ considers constraints that require that points from different classes are separated by a minimal distance after projection to the selected markers. Since all pairwise constraints would typically make the problem computationally too expensive, the constraints are sampled (`sampling_rate`) and capped (`n_neighbors`, `max_constraints`).

Figure 1: We consider the functional groups model defined in Table 1. In this model, a cell type is defined by its group (A_i, B_j, C_k) where $i = 1, 2, 3, 4$, $j = 1, 2$ and $k = 1, 2, 3, 4$. The gene expression G_s for $s = 1, 2, 3, 4, 5$ is drawn from a Gaussian distribution where the mean is determined by the type according to Table 1. (Left) we draw 20 cells from each of the 32 possible cell types. A random copy of each of the genes is drawn 100 consecutive times. In this example 5 markers are necessary and sufficient to distinguish all 32 classes. Note that even though all genes could technically be markers, the probability of selecting one from each class by selecting 5 genes at random is negligible. (Right) in this example we consider 10 cell types. The first 2 genes are repeated 10 times, the third gene 30 times, whereas the last two genes are repeated 150 times. The results of one-vs-all and scGeneFit for this synthetic data is shown in Figure 2.

- ‘pairwise_centers’ after projection to the selected markers every point is required to lie closest to its empirical center than every other class center (sampling and capping also apply here).
- **epsilon**: constraints will be of the form $\text{expr} > \Delta$, where Δ is chosen to be epsilon times the norm of the smallest constraint (default 1). This is the most important parameter in this problem, it determines the scale of the constraints, the rest the rest of the parameters only determine the size of the LP to adapt to limited computational resources. **We include a function that finds the optimal value of epsilon given a classifier and a training/test set. We provide an example of the optimization in https://github.com/solevillar/scGeneFit-python/blob/master/examples/scGeneFit_functional_groups.ipynb**
- **sampling_rate**: (if method==‘pairwise’ or ‘pairwise_centers’) selects constraints from a random sample of proportion sampling_rate (default 1)
- **n_neighbors**: (if method==‘pairwise’) chooses the constraints from n_neighbors nearest neighbors (default 3)
- **max_constraints**: maximum number of constraints to consider (default 1000)
- **redundancy**: (if method==‘centers’) in this case not all pairwise constraints are considered but just between centers of consecutive labels plus a random fraction of constraints given by redundancy. If redundancy==1 all constraints between pairs of centers are considered
- **verbose**: whether it prints information like size of the LP or elapsed time (default True)

References

- [1] Codeluppi, S. *et al.* Spatial organization of the somatosensory cortex revealed by cyclic smFISH. *bioRxiv* 276097 (2018).

Figure 2: **Small functional group simulation: Multivariate Normal Case.** We consider a toy dataset of 600 cells and 350 genes from 10 different cell types. The genes belong to one of the 5 different gene types described in Figure 1 (Right) and Table 1. **(Top left)** tSNE plot of the original data from Figure 1. Note that by construction all 10 classes are separated. A nearest centroid classifier on the original data has classification accuracy of 98%, indicating that the tSNE plot, even though it is a useful visualization tool, can actually be slightly misleading. We include this plot for visualization since it's standard in the field. The plots below show a more quantitative explanation of the results. **(Top middle left)** tSNE plot of the data restricted to the markers selected by the one-vs-all procedure. **(Top middle right)** tSNE plot of the data restricted to 10 markers selected by scGeneFit procedure. **(Top middle right)** tSNE plot of the data restricted to 18 markers selected by scGeneFit procedure. **(Second row):** scGeneFit performance improves significantly as the number of markers is allowed to increase: we plot the classification accuracy of a K-nearest neighbors classifiers as a function of the number of markers **(Third row):** Receiver operating characteristic curves for (left) one vs all, and (right) scGeneFit .

Figure 3: **Small functional group simulation: Poisson Link Case.** We consider a toy dataset of 600 cells and 350 genes from 10 different cell types. This model extends the multivariate normal functional group toy model in Figure 2 to include a Poisson link. The Poisson link has a strong effect on the performance of the one-vs-all analysis. In contrast scGeneFit is robust to this transformation. **(Top left):** a tSNE plot of the original data from Figure 1 followed by tSNE plot of the data restricted to the markers selected by the one-vs-all procedure. We observe that some classes are well distinguished but other classes are not. This is made quantitative in the second row plot. **(Top right):** a tSNE plot of the original data followed by tSNE plot of the data restricted to 9 markers selected by scGeneFit procedure. **(Second row):** scGeneFit performance improves significantly as the number of markers is allowed to increase: we plot the classification accuracy of a K-nearest neighbors classifiers as a function of the number of markers **(Third row)** Receiver operating characteristic curves for (left) one vs all, and (right) scGeneFit .

Figure 4: **Scaled-up functional group experiment: Multivariate Normal Case** We generated 40 cell labels from 15 different functional groups. We set each gene type to be repeated a random number of times between 10 and 1000. In this specific random example we have 1000 cells and 10,000 genes. Each run of the scGeneFit takes around 10 minutes in a Google Colab (standard account) (Top) tSNE plots of (from left to right) original data, data restricted to the 40 markers selected by one vs all, data restricted to the 40 markers selected by scGeneFit, data restricted to 45 markers selected by scGeneFit. (Bottom) We report the classification accuracy of scGeneFit and one vs all.

Figure 5: **Scaled-up functional group experiment: Poisson Link Case** We generated 40 classes from 15 different functional groups. We set each gene type to be repeated a random number of times between 10 and 1000. In this specific random example we have 1000 cells and 10,000 genes. Each run of the scGeneFit takes around 10 minutes in a Google Colab (standard account) (Top) tSNE plots of (from left to right) original data, data restricted to the 40 markers selected by one vs all, data restricted to the 40 markers selected by scGeneFit, data restricted to the 50 markers selected by scGeneFit. (Bottom) We report the classification accuracy of scGeneFit and one vs all.

- [2] Sladitschek, H. L. *et al.* Morphoseq: Full single-cell transcriptome dynamics up to gastrulation in a chordate. *Cell* (2020).
- [3] Zeisel, A. *et al.* Cell types in the mouse cortex and hippocampus revealed by single-cell RNA-seq. *Science* **347**, 1138–1142 (2015).
- [4] Baron, C. S. *et al.* Cell type purification by single-cell transcriptome-trained sorting. *CELL-D-19-01293* (2019).
- [5] Elyanow, R., Dumitrescu, B., Engelhardt, B. E. & Raphael, B. J. netnmf-sc: A network regularization algorithm for dimensionality reduction and imputation of single-cell expression data. In *RECOMB*, 297–298 (Springer, 2019).
- [6] Nelder, J. A. & Wedderburn, R. W. Generalized linear models. *Journal of the Royal Statistical Society: Series A (General)* **135**, 370–384 (1972).
- [7] Inouye, D. I., Yang, E., Allen, G. I. & Ravikumar, P. A review of multivariate distributions for count data derived from the poisson distribution. *Wiley Interdisciplinary Reviews: Computational Statistics* **9**, e1398 (2017).
- [8] Pierson, E. & Yau, C. Zifa: Dimensionality reduction for zero-inflated single-cell gene expression analysis. *Genome biology* **16**, 1–10 (2015).

REVIEWERS' COMMENTS

Reviewer #1 (Remarks to the Author):

The authors have addressed all my concerns through an impressive set of revisions.

Response to Reviews

1 Reviewer 1

The authors have addressed all my concerns through an impressive set of revisions.

Thank you for your thoughtful first review.